# A LITTLE LESS CONVERSATION, A LITTLE MORE ACTION, PLEASE: INVESTIGATING THE PHYSICAL COMMON-SENSE OF LLMS IN A 3D EMBODIED ENVIRONMENT

## ABSTRACT

As general-purpose tools, Large Language Models (LLMs) must often reason about everyday physical environments. In a question-and-answer capacity, understanding the interactions of physical objects may be necessary to give appropriate responses. Moreover, LLMs are increasingly used as reasoning engines in agentic systems, designing and controlling their action sequences. The vast majority of research has tackled this issue using static benchmarks, comprised of text or image-based questions about the physical world. However, these benchmarks do not capture the complexity and nuance of real-life physical processes. Here we advocate for a second, relatively unexplored, approach: 'embodying' the LLMs by granting them control of an agent within a 3D environment. We present the first embodied and cognitively meaningful evaluation of physical common-sense reasoning in LLMs. Our framework allows direct comparison of LLMs with other embodied agents, such as those based on Deep Reinforcement Learning, and human and non-human animals. We employ the Animal-AI (AAI) environment, a simulated 3D *virtual laboratory*, to study physical common-sense reasoning in LLMs. For this, we use the AAI Testbed, a suite of experiments that replicate laboratory studies with non-human animals, to study physical reasoning capabilities including distance estimation, tracking out-of-sight objects, and tool use. We demonstrate that state-of-the-art multi-modal models with no finetuning can complete this style of task, allowing meaningful comparison to the entrants of the 2019 Animal-AI Olympics competition and to human children. Our results show that LLMs are currently outperformed by human children on these tasks. We argue that this approach allows the study of physical reasoning using ecologically valid experiments drawn directly from cognitive science, improving the predictability and reliability of LLMs.

## 1 INTRODUCTION

Large Language Models (LLMs) can do your physics homework, but might not find their way to the classroom. While LLMs have made great strides in several areas, including writing code (Champa et al., 2024), solving maths problems (Frieder et al., 2024; Yuan et al., 2023b), and answering general knowledge questions (Wang et al., 2024), it remains unclear what they *know* and *understand* about the physical world.

Physical common-sense reasoning is the capacity to perceive, understand, and predict the behaviour of objects in an environment. This includes an understanding of the physical rules governing space and objects in that environment, and how they interact to determine the outcome of events or actions. In cognitive science, physical common-sense reasoning is also referred to as *intuitive* or *folk physics* (Kubricht et al., 2017). In LLMs, this capability has typically been evaluated using task- or image-based benchmarks involving short vignettes describing a physical scene, perhaps accompanied by an image if the model is multi-modal, with questions about the objects and their interactions (Buschoff et al., 2024; Bisk et al., 2020; Wang et al., 2023b). Benchmark scores are then aggregated to produce the final estimate of an LLM's capability. While this traditional approach has provided insight into

some aspects of physical reasoning, it misses many definitive features of physical *common sense* reasoning - that is, the capacity to *perceive, understand, and predict* the behaviour of objects in a physical environment, and use that knowledge to take appropriate actions.

Beyond this, traditional benchmarks suffer from a number of shortcomings (Hernández-Orallo, 2017). First, these benchmarks lack ecological validity—when deployed, LLM agents will not be interacting with well-described, clean vignettes with clear questions and uniquely identifiable answers. Instead, they will be interacting with a complex, noisy world where the correct answer, or action, is not always easily discriminated. Second, these benchmarks lack established construct validity (Borsboom et al., 2004; Cronbach & Meehl, 1955)—they have not been validated independently as *good* measures of physical common-sense reasoning by, for example, running experiments with humans or animals. Third, these benchmarks are static, meaning that the test items are fixed. When these benchmarks are released, there is a risk that new models will be trained on test items, contaminating the benchmark and thus rendering any results invalid, since models have been trained to predict the answer rather than to exhibit any emergent physical common-sense reasoning (Xu et al., 2024). Finally, benchmarks of physical common-sense reasoning are large and general—it is often unclear which *aspects* of physical common-sense reasoning they are targeting for evaluation. This is problematic because this type of reasoning is multifaceted, comprising everything from understanding inertia, gravity, and the solidity of objects, to reasoning about the concepts of causality, quantity and time (Lake et al., 2017; Shanahan et al., 2020). Traditional benchmarks do not allow us to precisely answer questions about what LLMs know about their physical environments *and* how they use that knowledge to take actions in them.

In this paper, we introduce *LLMs in Animal-AI* (LLM-AAI), a framework for conducting robust cognitive evaluations of the physical common-sense reasoning capabilities of LLM agents in a 3D virtual environment. Our framework allows us to test LLMs' physical common sense reasoning by embodying LLMs within Animal-AI—a *virtual laboratory* environment designed for the development of systematic cognitive test batteries with a particular emphasis on physical common-sense reasoning (Voudouris et al., 2023). Our approach situates LLMs in a physically realistic environment (ecologically valid), draws on testing materials that have been independently validated on humans and other animals (construct valid), capitalises on the variance of physical phenomena to produce difficult, dynamic tests (non-static), and tests a range of components of physical common-sense reasoning (precise evaluation target). A further strength of the LLM-AAI framework is that it facilitates comparison between human, animal, and multiple types of artificial intelligence systems on directly comparable tests. Here, we present the first evaluation of physical common-sense reasoning in LLMs using experiments drawn from research testing these capabilities in non-human animals, and compare their performance to Reinforcement Learning (RL) agents and human children.

The paper proceeds as follows: First, we review the recent literature on LLM agents and physical common-sense reasoning evaluations. Second, we introduce the Animal-AI environment and the Animal-AI Olympics—a competitive cognitive benchmark drawing on experiments from comparative psychology. Third, we introduce the LLM-AAI framework and describe the results from two experiments, where we evaluate the performance of three state-of-the-art LLMs (Claude Sonnet 3.5, GPT-4o, and Gemini 1.5 Pro) on the Animal-AI Olympics, in comparison to RL agents and human children, using different prompting strategies. Finally, we discuss these results and future work developing the LLM-AAI framework.

## 2 RELATED WORK

In machine learning and natural language processing, there is increasing interest in whether LLMs possess the capacity to perceive, understand, and predict the behaviour of objects in their environment, which has come to be known in the literature as *physical common-sense reasoning* (Bisk et al. 2020; Buschoff et al. 2024; Sap et al. 2020; Storks et al. 2019; Wang et al. 2023b; see also 'world models', e.g., Matsuo et al. 2022). This capacity has been studied extensively in the cognitive sciences, where it is often called *intuitive* or *folk physics* (Bates et al., 2019; Battaglia et al., 2012; Chiandetti & Vallortigara, 2011; Povinelli, 2003; Smith et al., 2018). Physical common-sense reasoning is multifaceted, ranging from understanding the properties and affordances of objects (Rutar et al., 2024) to tracking occluded objects (Voudouris et al., 2022b; 2024), using tools (Shanahan et al., 2020), and predicting the effects of gravity and momentum (Buschoff et al., 2024; Jassim

et al., 2024; Povinelli, 2003). One approach to studying physical common-sense reasoning in Large Language Models is through the administration of text-based descriptions of physical scenes, sometimes accompanied by images in the case of multi-modal LLMs, about which the model must answer some questions. The *Physical Interaction: Question Answering* (PIQA) benchmark (Bisk et al., 2020) consists of over 16K items that follows this approach using only text-based questions. LLMs are asked how they might achieve certain goals, such as *Make an outdoor pillow* and they are given two potential solutions, in this case, *Blow into a {trash bag, tin can} and tie with a rubber band*. Clearly, the answer is *trash bag*, given what we know as humans about the properties of trash bags and tin cans. Aroca-Ouellette et al. (2021) extend PIQA to over 18K question-answer pairs in the PROST benchmark, and Wang et al. (2023b) scale up even further to over 160K items in the NEW-TON benchmark. The results from these three benchmarks indicate that physical common-sense reasoning is not yet at human-level in text-only LLMs. In the multi-modal context, Buschoff et al. (2024) develop a suite of tasks inspired by cognitive science to study physical common-sense among other things. In their design, multi-modal prompts including task descriptions and visual stimuli are combined, and LLMs are tasked with providing a numerical judgment or rating about the described physical scene. For example, in the *block towers* task, LLMs are presented with pictures of stacks of coloured blocks, and asked to provide a binary judgment about whether the 'tower blocks' are stable or not. In their results, they found that only OpenAI's GPT-4V was able to make correct judgments above the level of chance on this task. In a similar vein, Jassim et al. (2024) present the *Grounding And Simulated Physics* (GRASP) benchmark, but in this case images are replaced with videos generated by a physics simulator. For every video, models are asked whether they think that the physical scene depicted is plausible, and they can only give a binary answer. Videos depict scenes in which objects appear to change size, colour, or shape spontaneously, disappear when occluded, or lack inertia or momentum. Their results also indicate that current LLMs that can process videos do not answer questions about these visual scenes above the level of chance.

An alternative approach to studying physical common-sense reasoning in LLMs is to grant them control of an agent, such that they are embodied in a real-world environment. Previous work has explored different approaches to LLM embodiment in both physical and digital environments. In the field of robotics, LLMs have been used to generate high-level action plans that are executed in real-world settings (Ahn et al., 2022; Driess et al., 2023; Jiang et al., 2022). However for such forms of deployment to be safe and reliable, it is important to establish whether LLM's apparent understanding of the physical world translates into appropriate behaviour when faced with real-world physical constraints (Ahn et al., 2022). Evaluating LLMs in 'real-world' contexts offers a high degree of ecological validity, but presents significant challenges: these approaches require extensive additional training, and face bottlenecks related to cost, safety and development speed in robotics. Hence, there is much to be gained from taking incremental steps towards true embodiment. One such step involves embedding LLMs as agents within virtual environments.

While there has been recent progress towards embodied LLM agents (Li et al., 2024), there has been no work, to our knowledge, on providing a robust framework for evaluating their physical common-sense reasoning. In the remainder of this section, we briefly review research on LLM agents before comparing it to our approach. LLM agents have been implemented and evaluated in a wide variety of game environments (Hu et al., 2024), ranging from co-operative games like *OverCooked* (Agashe et al., 2023; Gong et al., 2023; Liu et al., 2023; Zhang et al., 2023a) to strategy games like *StarCraft II* (Ma et al., 2023; Shao et al., 2024). Many of these games do not directly require good physical common-sense, because they involve simplistic visual and physical scenes with limited action spaces—their focus tends to be on evaluating how LLMs interact with other agents. In open field environments, there have been implementations of LLMs in Minecraft (Chen et al., 2024; Fan et al., 2022; Feng et al., 2023; Liu et al., 2023; Stengel-Eskin et al., 2024; Wang et al., 2023c;d;a; Yuan et al., 2023a; Zhang et al., 2023b; Zhao et al., 2024; Zhu et al., 2023) and Crafter (Du et al., 2023; Wu et al., 2024; Zhang et al., 2023c; Zhang & Lu, 2024), although again the physical reality of these environments is heavily limited by their simplicity - indeed, Crafter is a 2D world (Hafner, 2021). Most closely aligned to our work are those LLM implementations in VirtualHome (Huang et al., 2022; Xiang et al., 2024; Li et al., 2024), which has a realistic physics engine (Puig et al., 2018). In all cases, however, the focus has been on developing LLMs that can outperform humans or other AI agents, rather than developing a framework for more precise evaluation of physical common-sense reasoning.

This paper is the first example of a novel framework and proof-of-concept results demonstrating that LLMs can be evaluated on ecologically valid, complex tasks of physical common-sense reasoning. Furthermore, our approach allows meaningful direct comparisons to be drawn between LLMs and other agents, both biological (e.g. children) and non-biological (e.g. Reinforcement Learning agents).

## 3 THE ANIMAL-AI ENVIRONMENT

The Animal-AI (AAI) environment (Beyret et al., 2019; Crosby et al., 2019; Voudouris et al., 2023) is a physically realistic 3D simulation based on the Unity ML-Agents framework (Juliani, 2018), designed to be used by researchers from AI and cognitive science to assess nonverbal physical common sense reasoning in embodied agents. The goal of the environment is to offer a tool for interdisciplinary research at the intersection of AI and cognitive science, with a particular focus on comparative and developmental psychology. All experiments in AAI consist of a $40 \times 40$ arena, populated with a single agent (spherical with diameter 1) and a variety of different objects.

### 3.1 THE ANIMAL-AI TESTBED AND OLYMPICS

AAI was first released in 2019 as part of the Animal-AI Olympics Competition, in which over 60 entrants competed to produce agents that could solve a series of unseen tasks inspired by comparative psychology research (Crosby et al., 2020), thus favouring the development of agents that could perform robustly *out-of-distribution* on tests of physical common sense reasoning. After the competition was completed, these tasks were released as the Animal-AI Testbed to further stimulate interdisciplinary research between AI and comparative psychology. The Animal-AI Testbed contains 300 distinct tests (with 3 variants of each; n=900 tasks) that test the full breadth of capabilities that underpin physical common-sense reasoning, including navigating around obstacles, making spatial inferences, tracking occluded objects, and causal reasoning. The aim in every task is to maximise total reward at the end of the episode. The environment contains spheres of different colours and sizes: yellow spheres increase reward, as do green spheres, which also end the episode; red spheres decrease reward and end the episode. In all cases, the magnitude of the reward change is proportional to the size of the sphere. Touching red 'death zones' leads to a decrease in reward of $-1$ and also ends the episode. Reward decreases at a constant rate starting from 0 on each timestep, thus favouring efficient action sequences. Entering orange 'hot zones' leads to a doubling in reward decrement. A variety of movable and immovable blocks are present in the environment, including tunnels and opaque and transparent walls. While the colours and textures of objects in AAI are simplified, their physical interactions are close enough to those of the real world to appear identical. This is because AAI uses the physics engine provided in Unity: Every object has mass, volume, and static and dynamic friction coefficients, meaning that their movements are governed by laws of momentum, inertia, friction (including air resistance), and gravity.

The Animal-AI Testbed is arranged into 10 levels of 90 tasks of roughly increasing difficulty (Voudouris et al., 2022a) which probe different aspects of physical common-sense reasoning. For example, level 1 (*Food Retrieval*) tests the ability of the agent to navigate towards rewarding green and yellow spheres, level 2 (*Preferences*) tests the ability to distinguish objects that give different rewards, and level 3 (*Static Obstacles*) tests the ability to navigate around and over immovable solid objects, such as walls, ramps, and tunnels. The most complex levels test sophisticated physical common-sense reasoning abilities: level 8 (*Object Permanence and Working Memory*) tests whether agents understand that objects continue to exist when they are occluded, while level 10 (*Causal Reasoning*) tests the ability to understand cause and effect through the use of tools that can be used to achieve certain goals. These levels are described further in the Appendix in Table 1. Examples of the tests from each level used in this paper are presented in Figure 1.

## 4 METHODS

### 4.1 LLM-AAI

The LLM-AAI framework allows us to connect LLMs with the AAI environment. It is LLM-agnostic, requiring only a multimodal agent that can receive text-and-image inputs and return text

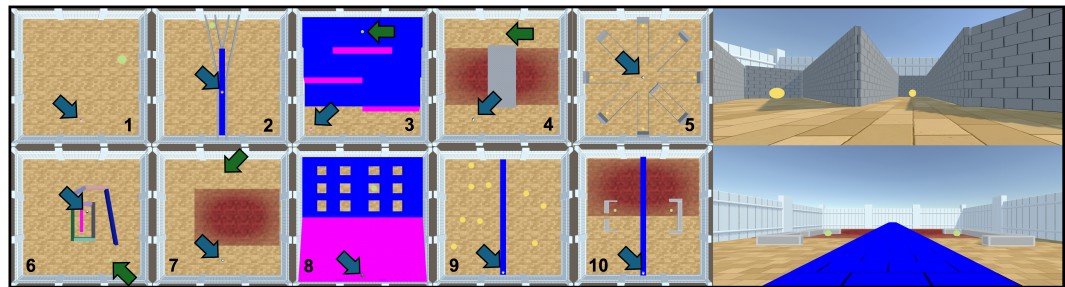

Figure 1: One task from each of the ten levels of the Animal-AI Testbed. The aim in every task is to collect as many yellow and/or green spheres while avoiding red zones, orange zones, and red spheres, before time runs out. Blue arrows indicate the location of the agent, and green arrows indicate the location of green spheres. The rightmost images show the agent's perspective during play in levels 5 and 10.

outputs. Figure 2 illustrates our approach. At each timestep, $t$, the environment returns a colour image of its current state, along with the agent's current reward and health. These observations are combined into a prompt and presented to the LLMs as a request.

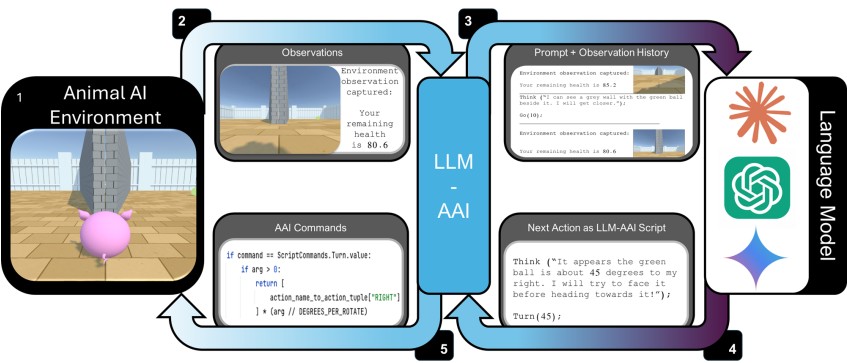

Figure 2: LLM-AAI. LLMs generate actions such as `Turn(45);` and passes them to LLM-AAI. LLM-AAI then parses these actions into commands that are understandable to the AAI environment and where they are subsequently executed. Observations from the environment are passed back to LLM-AAI, concatenated into the observation history, and provided, along with prompts like "Your remaining health is 80.6", to the LLM for reasoning and planning its next actions.

AAI requires an input on each frame describing how the agent should act (for example moving forwards or backwards, or rotating). We use an approach that finds a middle ground between requiring the LLM to provide such an input for each frame (which is costly), with approaches that require the LLM to interact with the environment by writing code that calls higher level APIs (Wang et al., 2023a) (which may outsource cognitively interesting tasks to specialised, environment-specific functions). LLMs act in the environment using a simple scripting language. The LLMs have access to three functions:

1. `Go`—this command moves the agent forwards (positive integer) or backwards (negative integer). `Go(1);` moves the agent one unit forwards, where the units are in the size of the agent. Due to the momentum of moving objects in the environment, higher values take the agent slightly further than the number of units specified. For instance, crossing the width of the arena can be achieved with the `Go(35);` command, even though the arena is 40×40 units.

2. `Turn`—this command rotates the agent right (positive integer) or left (negative integer). The units are in degrees of arc. `Turn(-90);` rotates the agent 90° to its left, while `Turn(90);` rotates the agent 90° to its right. In AAI, the minimum amount of rotation is 6°, so all values in the `Turn` command are rounded down to the nearest multiple of 6.

3. `Think`—the agent is instructed to use this command to describe the environment it observes, assess its position within that environment, track its remaining health and reward, and plan its course of action to collect the reward as efficiently as possible. For example, if the reward is behind the agent it might return `Think('I think the reward is directly behind me: I will turn around to look for it'); Turn(180);`. The inclusion of this command is influenced by approaches such as ReAct (Yao et al., 2022), in which LLM agents reason 'aloud'.

The LLM's response is parsed to return those scripts, which are converted into low-level action sequences, leading to a new state of the environment. Within a single episode, previous prompts and answers are prepended to the next prompt, so that the LLM has full access to previous states and action scripts. The LLM does not receive observations during the execution of action scripts.

## 4.2 LARGE LANGUAGE MODELS TESTED

We consider three state-of-the-art multi-modal Large Language Models. Our selection was based on a convenience sample, guided by the inclusion criterion that models must be multi-modal with a large context window (>64k), and the exclusion criterion that models must not be too costly to run inference on. We evaluated **Claude 3.5 Sonnet**, **GPT-4o**, and **Gemini 1.5 Pro**. We ran all experiments with temperature 0, but noticed that model responses can vary nevertheless. Therefore, we ran three trials of each model on each task.

## 4.3 EXPERIMENTS

In this study, we use a subset of the Animal-AI Testbed containing four randomly selected tasks from the ten levels (n=40), replicating the design of Voudouris et al. (2022a), in which 59 children aged 6-10 completed the same subset of 40 tasks. This allows direct comparison of LLM agents with human children, and non-human entrants to the Animal-AI Olympics Competition (Crosby et al., 2020).

We conduct two experiments to explore LLM performance in this setting. Our first experiment includes a prompt that explains the environment and possible actions to the LLM, and assesses three models on 40 AAI Testbed tasks. Our second experiment provides the LLM with an in-context example of the completion of a simple 'tutorial' level, which we assess on a subset of the 40 instances assessed in Experiment 1.

When we encountered errors from API calls that persisted after three retries, we discarded the current trial data and relaunched that trial run.

## 4.4 EXPERIMENT 1: REACT PROMPTING

First, we designed a simple prompt that provides the core information needed to navigate and collect rewards in the AAI Testbed. To improve the LLM's decision-making, we incorporated the ReAct (Reasoning and Acting) framework (Yao et al., 2022) into our prompt design. The ReAct approach combines reasoning and acting by allowing the model to generate reasoning traces alongside actions, which can improve performance on agentic tasks (Yao et al., 2022). By integrating ReAct, we encourage the LLM to first reason about the environment—identifying visible objects and their spatial relationships relative to the agent—before producing action scripts.

Our prompt begins by setting the context: The LLM is informed that it is a player in a game set in a square arena with a white fence, tasked with collecting green and yellow ball rewards as quickly and efficiently as possible using a basic scripting language. The prompt details the kinds of objects the LLM will encounter, their key properties, and instructions on how to write scripts using the commands `Think`, `Go`, and `Turn`. It includes examples to illustrate correct usage of these commands and provides guidelines to avoid common mistakes.

To aid the LLMs in navigating the environment efficiently, we incorporated expert tips on movement distances and turning angles. For instance, we explain that moves of 1 to 10 steps cover small distances, while moves of 10 to 20 steps cover larger distances. We also provide strategic guidance on

how to approach the task using the `Think` command to describe the current state of the environment and plan its actions, and subsequently using either `Go` or `Turn` to move within the environment.

Lastly, the prompt warns about potential obstacles such as red lava puddles, holes, blue paths, purple ramps, transparent walls, pushable grey blocks, and immovable objects like walls and arches. It provides instructions on how to identify and interact with these obstacles, emphasizing caution to prevent the agent from dying or becoming trapped. The full prompt is provided in Appendix C.

Armed with this prompt, each LLM is evaluated on the 40 tasks performed by children in Voudouris et al. (Voudouris et al., 2022a). The LLM is not presented with previous action scripts from other episodes, meaning it approaches each task as if it is interacting with the AAI Testbed for the first time.

### 4.5 EXPERIMENT 2: SUPERVISED IN-CONTEXT LEARNING

When children played the tasks in the AAI Testbed, they received a short two-minute video to describe "the game"—that is, to introduce the AAI environment, its objects and controls. To emulate this, we designed an example level in AAI that introduced the same information presented in the video, including a sequence of scripts for solving the level and 'Think' actions to explain observations. These were incorporated into the prompt above. LLMs are thus provided with images of objects they may encounter, as opposed to just textual descriptions, and an 'expert example' (shown in Appendix D), before they are tasked with controlling the agent. We call this *supervised in-context learning*.

Due to the increased cost of passing additional images and text, we conducted this experiment on a subset of tasks. We focused on the first three levels of the AAI Testbed as they provide a better opportunity to observe meaningful differences given LLMs' poor performance on later levels in Experiment 1.

## 5 RESULTS

### 5.1 EXPERIMENT 1

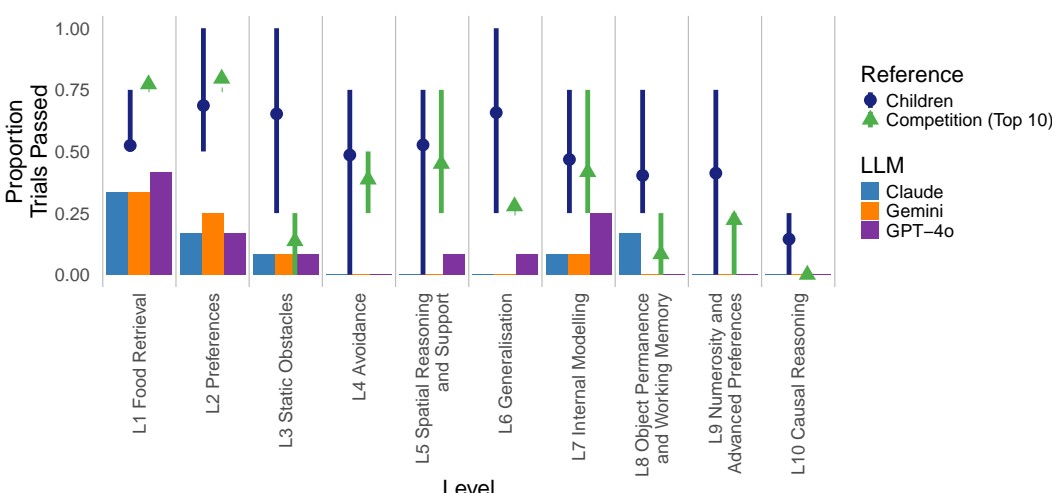

Figure 3: The proportion of trials passed by each LLM on each level, consisting of 3 trials of 4 tasks each (total n=12 trials per level). The interquartile range of proportions for all children (n=59) and the top 10 entrants to the Animal-AI Olympics Competition are presented as bars, with overall proportion for those populations indicated by points. Note that the children and competition agents have error bars, while the LLMs do not. This is because the child and competition agents contain a population of different individuals, across which we would like to understand variation, while the LLMs are repetitions of the same individual, and so are aggregated into a single value.

Our results, summarised in Figure 3, show that LLMs are able to complete some challenges in Levels 1 and 2, with sporadic performance across Levels 5, 6 and 8. They are comparable in performance with competition agents in Levels 3, 8, 9 and 10, however these all occur at a very low success rate, so there may be a floor effect obscuring a difference in capability between the groups. Children perform convincingly better than LLM agents across all levels, with child error bars only overlapping with LLM performance in Levels 4, 5, 9 and 10, where LLM performance is very low.

These results show that LLMs are able to perform successfully in the simplest tasks of the testbed, but performance drops off quickly in more challenging tasks. The LLMs' performance never exceeds that of the top 10 agents submitted to the Animal-AI competition. It could be argued that this comparison will always favour the RL agents, who had been specifically trained for the environment, if not for the specific tasks. However, the same cannot be said for the human children, whose performance also exceeded that of the LLMs across the board. These results indicate that LLMs may still lack the physical common-sense reasoning abilities possessed by human children.

## 5.2 EXPERIMENT 2

The supervised in-context learning results are shown in Figure 4. Each LLM's performance is illustrated by a pair of bars. The first bar illustrates performance *without* our 'expert example', and is identical to the results of the main experiment from Figure 3, while the second bar represents performance *with* our example and is new in the in-context learning experiment.

Overall, we did not observe a notable difference in performance when providing the LLMs with the 'expert example'. While the LLMs still broadly perform successfully on these early levels, they do not outperform the competition agents or the children.

The observed performance difference, when including the 'expert example', was not the same across all the tested LLMs. Claude performed slightly worse in Level 1 than it had without in-context learning, whereas the opposite occurred in Level 2. Performance on Level 3 stayed the same. For Gemini, the addition of in-context learning had either no effect, in Level 1, or decreased the proportion of trials passed, in Levels 2 and 3. While GPT also experienced no performance difference in Level 1, its results rose both in Levels 2 and 3, with its Level 3 proportion of trials passed matching the upper interquartile range of the competition agents and the lower range of the children.

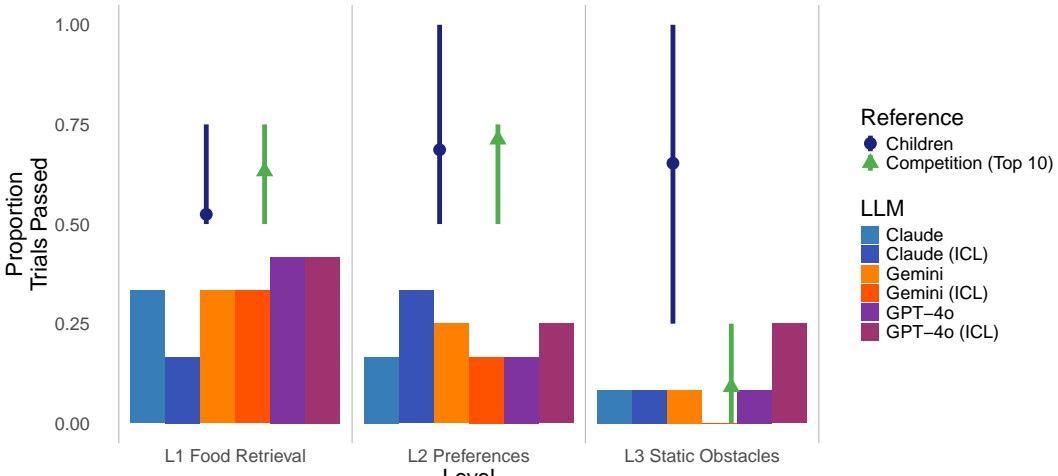

Figure 4: The proportion of trials by each LLM on each level, consisting of 3 trials of 4 tasks each (total n=12 trials per level). The interquartile range of proportions for all children (n = 59) and the top 10 entrants to the Animal-AI Olympics Competition are presented as bars, with overall proportion for those populations indicated by points.

## 6 DISCUSSION

The LLM-AAI framework tests the *out of the box* physical reasoning capabilities of LLMs by allowing them to perceive and interact with the Animal-AI environment via the ReAct prompting method (Yao et al., 2022). While previous work has explored the capabilities of LLMs in virtual environments, none have used them to develop a framework for testing physical common-sense reasoning in LLMs. Our results show that this method LLMs can not only be assessed in this way, but that when this is done it allows meaningful comparisons to be made with other biological and non-biological intelligences.

Evaluations in LLM-AAI have synergies with other efforts in evaluating and training LLMs. In evaluation, several LLM testbeds can be seen as targeting facets of the Animal-AI Testbed such as spatial reasoning (Ranasinghe et al., 2024), numerosity (Trott et al., 2017; Villa et al., 2023) and tool use (Tian et al., 2023). Evaluations in LLM-AAI complement such efforts, but also adds the increased challenge of interacting in a 3D environment, which has less direct correspondence with the linguistic prompt. Furthermore, where a 3D environment has been used at the learning stage (Dagan et al., 2023; Zellers et al., 2021; Driess et al., 2023; Xiang et al., 2024), an LLM-AAI approach can be used to ensure the robustness of a model's physical common-sense.

For humans, an understanding of the physical world is built from countless embodied interactions with objects in their environment (Thelen, 2000). It is from these interactions that humans construct intuitive theories of the causal relationships that exist in their external world (Goddu & Gopnik, 2024; Gopnik & Schulz, 2004; Tenenbaum et al., 2011), and ground the symbolic concepts contained in language (Lakoff & Johnson, 2008; Wolff, 2007). To date, there has been much debate as to the potential for 'disembodied' systems such as LLMs to have a 'meaningful' understanding of the physical world, or even a 'world model' (Bender & Koller, 2020; Mitchell, 2021; Shanahan, 2010). The LLM-AAI framework allows us to make headway on these debates, with our initial results suggesting that LLMs still have some way to go before they can compete with their embodied counterparts.

### 6.1 LIMITATIONS AND FUTURE WORK

The LLM-AAI framework satisfies an important demand in the field of LLM evaluation. It provides a methodology and way forward for evaluations of physical common-sense reasoning using independently developed tests from cognitive science (construct valid) that measure specific components of physical common-sense (precise evaluation target), in a physically realistic environment (ecologically valid) with real-world dynamics (non-static). These tests identify the capabilities and failure modes of contemporary multi-modal LLMs, aiding researchers to identify how training curricula and model architectures can be improved to achieve better performance. Furthermore, LLM-AAI enables direct, cognitively meaningful, comparisons between LLMs, deep reinforcement learning (DRL) agents, humans, and other animals. Our results in this paper demonstrate that out-of-the-box systems can produce meaningful results on the Animal-AI competition. Nevertheless, there remain a number of extensions to how LLMs interact with AAI through our framework that could improve LLM performance. These extensions remedy some of the limitations of this current work and serve as the basis for future research.

**Sensing the environment.** In LLM-AAI, at every conversation turn, the tested LLM receives a single 512 x 512-pixel image of the environment. This image is captured after the LLM's action script is executed. The number of environment time-steps that unfold during the execution depends on the action script. For example, if the LLM uses the `Turn(180)` command, more environment time-steps will go by than if the LLM uses the `Turn(25)` command. Despite this difference in time-steps, in both cases a single image observation is sent to the LLM. While this observation routine allows larger agent-displacements with fewer API calls (and hence reduced costs), it can also cause the LLM to miss important environment information. For example, the agent may execute a `Turn(180)` script meaning that it misses the goal that is placed 90° to its right.

**Locomotion and control.** The control scheme used in the study, although theoretically sufficient for completing levels, is a relatively coarse way of controlling an agent in the environment compared to both children and AAI Olympics competition entrants, who could all provide a single action after every timestep. The additional challenge of writing action scripts manifests in the game-play of

the LLMs. For example, in many cases, the LLM almost aligns itself with the goal but misses it slightly. This could result in the LLM finding itself beyond the goal and having to take extra turns to reorient itself before trying again. Future work could experiment with alternatives to the control scheme employed in this paper, such as allowing the LLM to control the agent frame-by-frame, or fine tuning a model to turn natural language descriptions of the action into environment commands.

**Capability limitations.** This study aimed to assess LLMs *out of the box* on the Animal-AI Testbed. This ensures that the evaluation is not contaminated as LLMs have not been explicitly trained to solve these tests. However, it might be that the challenge of controlling the agent in the environment is so large that this dominates the cognitive challenge on some tasks. By comparing the LLMs' `Think` responses with their in-world actions on selected levels (see Capability Case Studies in Appendix A), we describe a specific example (object permanence) where low-level navigational demands may have limited LLMs' performance, among other indicative failures (in affordance understanding and numerical magnitude comparison) that may shed light on the the behavioural mechanisms underlying our results. Future work could ensure LLM performance is not constrained by low-level perceptual or navigational demands by fine-tuning multi-modal LLMs on the observations and action scripts of an agent successfully completing simple navigation tasks. This would overcome the problem of calibrating action scripts to the environment, and allow our tests to more accurately reveal the cognitive capabilities of LLMs. An alternative approach would be to embed LLMs as components of a larger control and memory system (Wang et al., 2023a; Sumers et al., 2023) to attempt achieve better performance on the Animal-AI Testbed.

**Cost.** The scaling cost of longer experiments rendered some experiments financially unfeasible. For example, human participants completing the same tasks as the LLM would have had the ability to learn over the course of the 40 arenas; this could be replicated in LLMs by attempting all 40 arenas in a single context window. However, the large number of tokens this generates is too costly. Due to financial limitations, the tested LLMs were also restricted to using, at most, 30 action-scripts, and therefore API calls, per episode. In contrast, human participants and DRL agents were only restricted by the arena's time-limit, rather than a maximum number of executed actions. This constraint was especially penalising for LLMs in arenas with multiple goals and those that required many finely controlled movements and adjustments; such sequences inflated the number of action-scripts needed to complete the level. Future work will increase or remove the action-script limit and assess the change in performance.

**Towards cognitively-driven evaluation.** The levels in the Animal-AI Testbed are inspired by the rich tradition of developing non-verbal tests of capacities in cognitive science. Since there exists a large number of tests and experimental paradigms, they cannot be condensed into a single testbed such as ours. More targeted LLM-AAI evaluations using the tests from Voudouris et al. (2022b) for object permanence or Rutar et al. (2024) for object affordances, will allow assessors to make more precise statements about physical common-sense reasoning capabilities, and produce comparisons with the humans and DRL agents that have been evaluated on these tests.

## 7 CONCLUSION

We have introduced LLM-AAI, a framework for evaluating the physical common-sense reasoning capabilities of LLMs in a 3D environment. Using the diverse tasks of the Animal-AI Testbed, we have presented results from an initial assessment, showing that LLMs are capable of completing tasks using LLM-AAI, but may lack the physical common-sense reasoning capabilities of humans. We hope that these results will inspire researchers to embrace embodied evaluations as a powerful addition to the LLM evaluation toolbox.

## 8 REPRODUCIBILITY STATEMENT

All the results presented in this paper can be reproduced, provided that the closed-source LLM checkpoints that were tested are not altered. The checkpoints used were:

- Claude 3.5 Sonnet: claude-3-5-sonnet-20240620
- GPT-4o: gpt-4o-2024-05-13

- Gemini 1.5 Pro: gemini-1.5-pro-001

During our experiments we encountered issues with the API for Gemini 1.5 Pro, these issues were the only occasions in which we had to discard and rerun trials, as it stopped us from collecting complete data for trials. The API issue we encountered is documented at https://github.com/google-gemini/generative-ai-python/issues/559.

We also make the prompts that were passed to the LLMs available in Appendices C and D. We produced all of our results using Animal-AI version 3.1.3. Source code for our experiments is available at https://github.com/Kinds-of-Intelligence-CFI/LLM-AAI.

## 9 ETHICS STATEMENT

No human or animal participants were involved in this study, and no sensitive topics were used or contained in our interactions with LLMs. The human data used in our comparison was from an openly available dataset from an independent study found here: https://osf.io/g8u26/.

## 10 ACKNOWLEDGMENTS

This work was partly funded under the Kinds of Intelligence project, The Leverhulme Centre for the Future of Intelligence (RC-2015-067), and an ESRC scholarship to BS (ES/P000738/1).

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

# A   CAPABILITY CASE STUDIES

Beyond the overall pass-rates of models, our evaluations in LLM-AAI also generated a rich dataset of behaviours and `Think` actions which can be used to investigate the reasons for LLM performance further. In this section we assess LLM performance in some cognitive domains.

## A.1   AFFORDANCE UNDERSTANDING

Affordance understanding is "the cognitive capability to identify what action-possibilities exist with a particular object or set of objects, given an agent's specific physical properties and capacities" (Rutar et al., 2024). Our results demonstrate interesting failures of affordance understanding in LLM agents.

In arena '10-22-03' of the AAI testbed the reward is on a platform. To reach the reward the agent must push a block to bridge the gap between the platform and a ramp that faces it, and then climb the ramp [1]. To do this, the agent must have an understanding of two sets of affordances: That certain blocks can be pushed, and that ramps can be climbed from a certain side.

No models considered using the pushable block, but only GPT-4o consistently noted its existence, indicating that for the others their vision may not have been sensitive enough to detect it. However, all LLMs acknowledged the existence of the ramp. For example Gemini 1.5 Pro stated 'I see a purple ramp to my right and the blue path is still visible'. Of the three models tested only Claude Sonnet 3.5 noticed the ramp and attempted to climb it, for example stating 'I need to climb this ramp to explore what might be on the other side'. However, it did not consider the fact that ramps must be climbed from a particular side and so failed to climb it. The fact that LLMs were able to recognise the ramp, but only one realised its affordance of being climbable, without realising that this affordance is only available from one side, indicates that robust affordance understanding is still a significant challenge for models.

## A.2   OBJECT PERMANENCE

Object permanence is "the understanding and belief that objects continue to exist even when they are not directly observable" (Voudouris et al., 2022b), and the presence of this capability is a foundational milestone in human cognitive development (Piaget, 2013; Baillargeon et al., 1985). Although LLM performance on object permanence tasks was generally poor (see Figure 3, Level 8), verbal report from the models suggests that these failures may be due to low-level navigational or perceptual difficulties, rather than failures of object permanence *per se*.

In arena '08-03-03', two yellow rewards descend from above before being hidden behind a series of walls on the other side of the arena. To solve this task, agents must reason that although the rewards are no longer visible, they nonetheless *continue to exist*, and can be discovered by searching for them behind the wall.

All LLMs reported that they were searching for the rewards that they had seen previously, while GPT-4o and Claude Sonnet 3.5 made explicit comments relating to the continual existence of the rewards despite them no longer being in view. For example, GPT-4o states: 'I can no longer see the yellow balls. They might be behind the grey blocks ahead. I will turn to the right to get a better view.' And then, after series of poorly executed actions, GPT-4o continues: 'I still cannot see the yellow balls. They must be behind the grey blocks. I will ... move closer to investigate.'

In contrast to the other models, Claude Sonnet 3.5 showed some success retrieving the reward (see Figure 3, Level 8) and their verbal reports also suggest a coherent strategy. At the start, Claude comments: 'There appears to be grey block structures in front of me, which might be obscuring the view of the balls.' After moving closer, Claude continues: 'It seems the balls might be behind these structures. I need to move forward and to the right to try to get around these obstacles and locate the yellow balls.'

Given that these verbalisations are being provided in response to dynamic visual input in an embodied environment, rather than as part of a purely linguistic interaction, they make a more robust case

---

[1]An alternative solution involves building up momentum on the ramp to jump the gap. This solution was not discovered by any agents.

for the presence of a generalisable object permanence capability that future work could investigate systematically.

## A.3 NUMERICAL MAGNITUDE COMPARISON

Numerical magnitude comparison is the ability to determine which one of two numbers has the greater magnitude.

In our experiments, failures in numerical magnitude comparison arose when the LLMs attempted to track the change in their health. In LLM-AAI, at every turn, before the LLM agent provides a new action script, the environment states the agent's current health value. An example of this might be: 'Your remaining health is 83.4', which is passed as user content to the LLM assistant. The agent may then infer, from this health value, whether it has collected a reward while executing its last action script, by comparing the value with the one it was told one turn before. Misjudging this difference in health may lead to misinterpreting whether or not a reward has been collected.

All three tested LLMs showcased occasional errors in comparing previous and current health values. The following example illustrates the most common flow in which this issue was observed. In arena '05-09-01', GPT-4o attempts to collect a yellow reward in its view. The LLM is provided with a health reading of 63.3 followed by one of 59.7. Clearly, the health has decreased as the agent has not collected the reward. Surprisingly, however, its following `Think` command content—'My health has increased, confirming the collection'—showcases an inability to correctly compare the numbers 63.3 and 59.7. Similar inaccuracies were observed for Claude Sonnet 3.5 and Gemini 1.5 Pro. In one example, Claude Sonnet 3.5 explicitly verbalised that a health decrease was an increase: In arena '04-16-01', after missing the green reward, Claude stated 'I have successfully collected the green ball as my health has increased from 84.2 to 35.4'. This rarer example illustrates how numerical mistakes may also lead to the LLM forgetting some basic rules of the environment. Namely, that if it *had* collected the green reward, the episode would have ended.

Our goal was not to conduct a statistical study of the occurrence of this failure mode or to compare numerical magnitude comparison in different LLMs. Rather, we demonstrate that this ability can be crucial to completing physical common sense tasks.

## B  THE ANIMAL-AI TESTBED

The Animal-AI Testbed contains 10 levels of 30 tasks with 3 variants each (n=900 tasks). Each level tests different aspects of physical common-sense reasoning. A description of each level is presented in Table 1 overleaf. Participants in the Animal-AI Olympics Competition were tested on all 900 tasks of the Testbed, and developers were not given access to the contents of the Testbed prior to submission to the competition. In our plots in Section 5, we only report the top 10 entrants to the competition in terms of overall score, indicating the current best performance of deep reinforcement learning (DRL) agents tested out-of-distribution. Data from children (n=59) on 4 tasks from each of the 10 levels (n=40) were taken from Voudouris et al. (2022a). All comparisons between LLMs, children, and competition agents is based on their performances on only these 40 tasks. In the Animal-AI Testbed, objects with specific functions have fixed colours. Ramps are always purple, platforms are always blue, and pushable blocks are always light grey. Other blocks may take any colour.

Table 1: The Animal-AI Testbed consists of 10 levels of 30 tests with 3 variants each (n=900 tasks). Each level tests a different aspect of physical common sense reasoning.

| Level | Description |
| --- | --- |
| L1 - Food Retrieval | Navigation towards rewarding objects in a large arena. |
| L2 - Preferences | Choice between objects with different reward values, indicated by their size and colour. |
| L3 - Static Obstacles | Objects partially occluded behind static obstacles around which agents must navigate. |
| L4 - Avoidance | Navigation around punishing objects to obtain rewarding objects. |
| L5 - Spatial Reasoning and Support | Based on the absence of rewarding objects in one part of the arena, agents must infer their presence elsewhere, even when (partially) occluded. Rewarding objects may also be out of reach on a ledge or a pillar, requiring the agent to push a movable block to knock them down. |
| L6 - Generalisation | Tasks from the first five levels are adapted so that the surroundings are colours. |
| L7 - Internal Modelling | Tasks from the first five levels with alternating periods in which visual information is withheld, as though the lights have gone out. The agent must continue to model their environment during these periods. |
| L8 - Object Permanence and Working Memory | Rewards are hidden behind obstacles for the agent to find. |
| L9 - Numerosity and Advanced Preferences | Discrimination between different numbers of rewarding objects, testing the ability to visually discriminate and count objects in a scene. |
| L10 - Causal Reasoning | Rewarding objects are only obtainable using a tool with certain physical properties and affordances. |

## C   INITIAL PROMPT

You are a PLAYER in a game set in a square arena with a white fence. Your
  task is to collect all the rewards as quickly and efficiently as
  possible using a basic scripting language. The rewards are green and
  yellow balls.

To successfully collect a reward, you must fully pass through it. For
  example, if you think the reward is 10 steps away, you should go
  further than 10 steps to ensure you collect it, e.g., Go(15);.

The game ends when you have collected all the rewards and the arena
  closes. If you are still in the arena, the game is NOT finished and
  you have NOT collected all the rewards.

Your remaining health is displayed in the environment as "Your remaining
  health is:". The game will end if your health reaches 0.

NOTE: When you collect a reward, your remaining health will INCREASE
  compared to the previous timestep. If it doesn\'t increase, the
  reward was not collected. Always compare your current health with the
   previous timestep to confirm this. The scripting language consists
  of commands in the form <COMMAND>(<ARG>);

Note:
– If ARG is numerical it should always be an integer, never a float.
– DO NOT include any response not following the format of the scripting
  language. Doing so will result in failure.
– DO NOT wrap your commands with inverted commas: \' \'Think(\'Something
  \');\'Go(5);\' \' would fail whereas \' Think(\'Something\');Go(5);
  \' would not.

Commands are:

– Think: Reason about what actions to take to collect the rewards most
  efficiently (does not affect the environment). Note: Always format
  the thought as a string. Also, when using this command, do not
  include parentheses as arguments. For example, correct: \'Think(\'I
  cannot see the reward———yellow or green ball———in the arena\');\'
  Incorrect: \'Think(\'I cannot see the reward (yellow or green ball)
  in the arena\');\'

– Go: Move forward or backward a certain number of steps (1 to 35 steps
  forward, –1 to –35 backward).

– Turn: Turn by a specified number of degrees (any positive number
  between 1 and 360 degrees turns the character to the right (clockwise
  ) and any negative number between –1 and –360 degrees turns the
  character to the left (anticlockwise)).

Examples:
To move forward by 5 steps: \'Go(5);\'.
To investigate what is happening to your left: \'Think(\'I would like to
  investigate what is happening to my left\');Turn(-90);\'

The number of scripts you can send is limited, so try to complete the
  levels efficiently.
The size of the arena is 35 by 35: \'Go(35)\' will take you from one end
  of the arena to the other.
After you submit your script, you will receive an image observation. Use
  this image to plan your next script.

EXPERT TIPS:
– Moves of 1 to 10 steps cover small distances, while moves of 10 to 20
  cover larger distances.

- Turns of 25 to 45 degrees turn you a small amount to the right, while turns of −25 to −45 degrees will turn you a small amount to the left. DO NOT use turns less than 25 degrees.
- Turns of 45 to 90 degrees will turn you a large amount to the right, while turns of −45 to −90 degrees will turn you a large amount to the left.\n
- Turning 180 or −180 degrees will turn you all the way round so that you are facing backwards.

How to approach the task:

Start by using the \'Think\' command to describe the environment you see. When you find the rewards, i.e. green or yellow balls, ALWAYS explicitly state BOTH your DISTANCE and ANGLE with respect to them. Note: Only green and yellow balls are rewards and nothing else.
Take appropriate actions. Use \'Go\' OR \'Turn\', but DO NOT combine them in the same turn. Always follow \'Think\' with one of these two actions.

HINT: Your vision is good but not perfect and some rewards may not be immediately visible. Rewards may be behind you. Explore the arena to locate them. When exploring, try to get a 360-view of the arena. If both green and yellow balls are present, collect the yellow balls first and green balls last. Note that some arenas may not have green balls at all. The reward you get is proportional to the size of the ball: make sure to get the bigger balls first!. Finally, the lights may go out during a level. They may or may not come back on: use what you\'ve learnt about the arena so far to move around and collect the reward when this happens!

When you find a reward:
Use the \'Turn\' command to align yourself directly with the reward. Before moving towards it, check the observation image provided by the environment to ensure the reward is centered in your view. If the reward is not centered, adjust your alignment with additional turns until it is.
Use the \'Go\' command to move toward the reward.
If the reward is more than 15 steps away, align yourself with the reward as best as you can and move half the distance first. Then reassess your angle with respect to the reward, use \'Turn\' to adjust your angle if the reward is not centered in your view, and move the remaining distance.
Remember: ALWAYS check your health after collecting a reward. You have successfully collected the reward only if your health has INCREASED compared to the previous timestep.

Be mindful of obstacles:

Red lava puddles and red balls: If you run into them, you will die.
Holes: Some may contain rewards, but if you fall into an empty hole, you will be trapped and unable to collect other rewards.
Blue paths: These are slightly raised paths. You can walk on them, but once you step off, you won\'t be able to get back onto them.
Purple ramps: You can climb them to get to the other side. Once you climb over the ramp, you cannot climb back over the same ramp.
Transparent walls: You can see through them, but you cannot walk through them.
Pushable grey blocks: These are cube-like structures, patterned with dark grey rectangles on each face. If viewed from one side, they will look like a rectangular structure. They can be pushed, but they are heavy! To move these blocks, you need to run into them. The blocks are heavy so you need to add extra steps to your Go() command.
Immovable objects: Walls and arches cannot be moved.
Ready to play? You will start by seeing three image observations.
A new level begins now. Environment observation captured

## D    ICL PROMPT

In the additional, in-context learning, experiment, the initial prompt was accompanied by a demonstration of an episode, which included examples of objects it may encounter in AAI. We replicate this below in human-readable format; with observations in sequence, and their responses below:

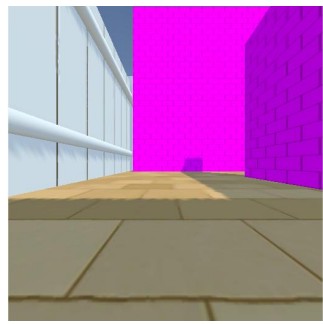

Figure 5: ⟨ Initial image: no response ⟩

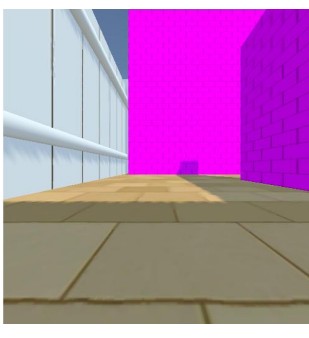

Figure 6: ⟨ Initial image: no response ⟩

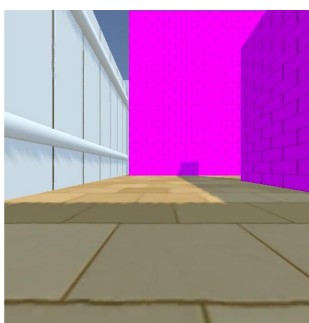

Figure 7: Think("I can move forwards and backwards with Go");Go(3);

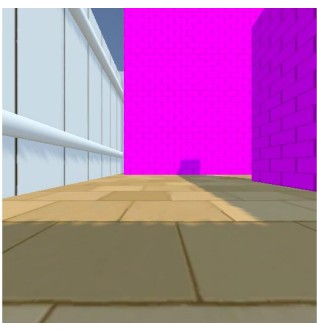

Figure 8: Go(-3);

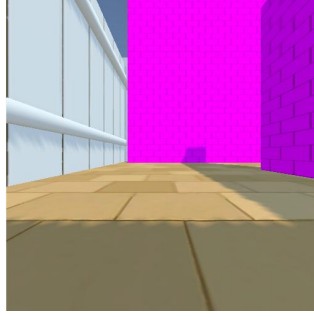

Figure 9: Think("I can turn left and right with Turn");Turn(30);

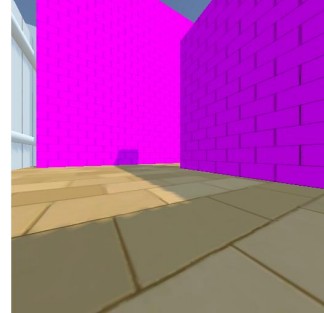

Figure 10: Turn(-30);

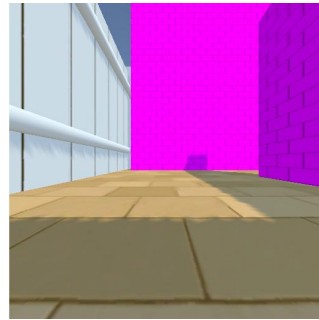

Figure 11: Think("I'll move through this level and take a look at some of the items I might encounter");Go(9);

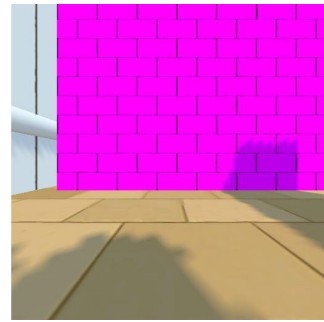

Figure 12: Turn(90);

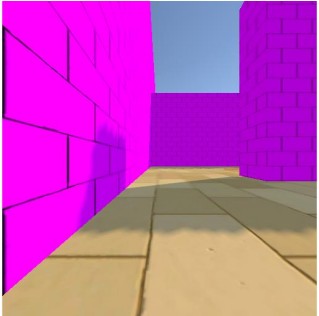

Figure 13: Go(9);

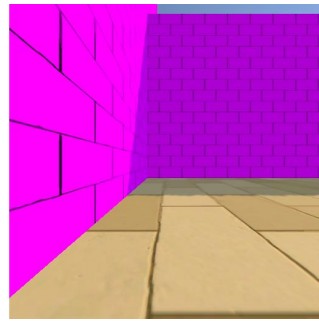

Figure 14: Turn(90);

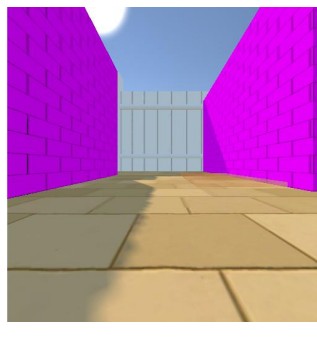

Figure 15: Think("I shouldn't stand on the orange zone for too long. I'll lose health more quickly!");Go(10);

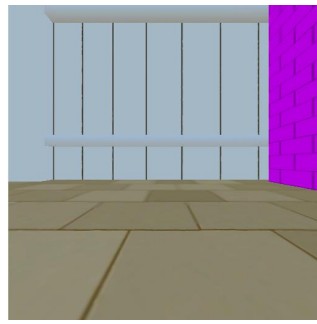

Figure 16: Turn(-90);

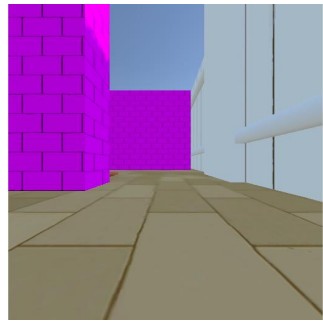

Figure 17: Go(9);

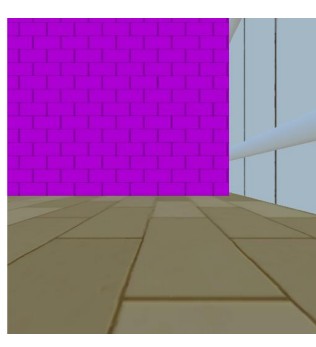

Figure 18: Turn(-90);

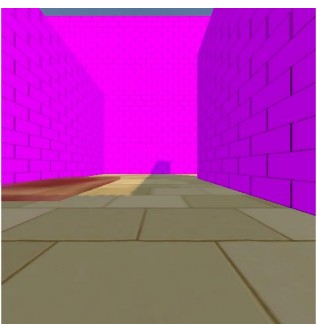

Figure 19: Think("I shouldn't step in the red zone at all: that will make me lose the level!");Go(9);

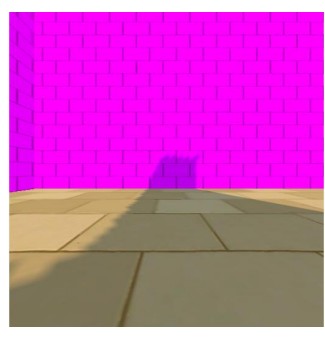

Figure 20: Turn(90);

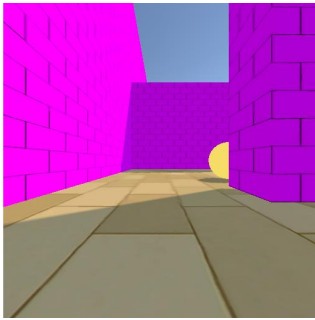

Figure 21: Go(7);

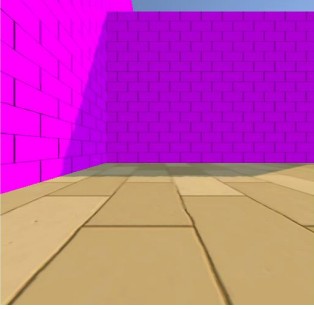

Figure 22: Turn(90);

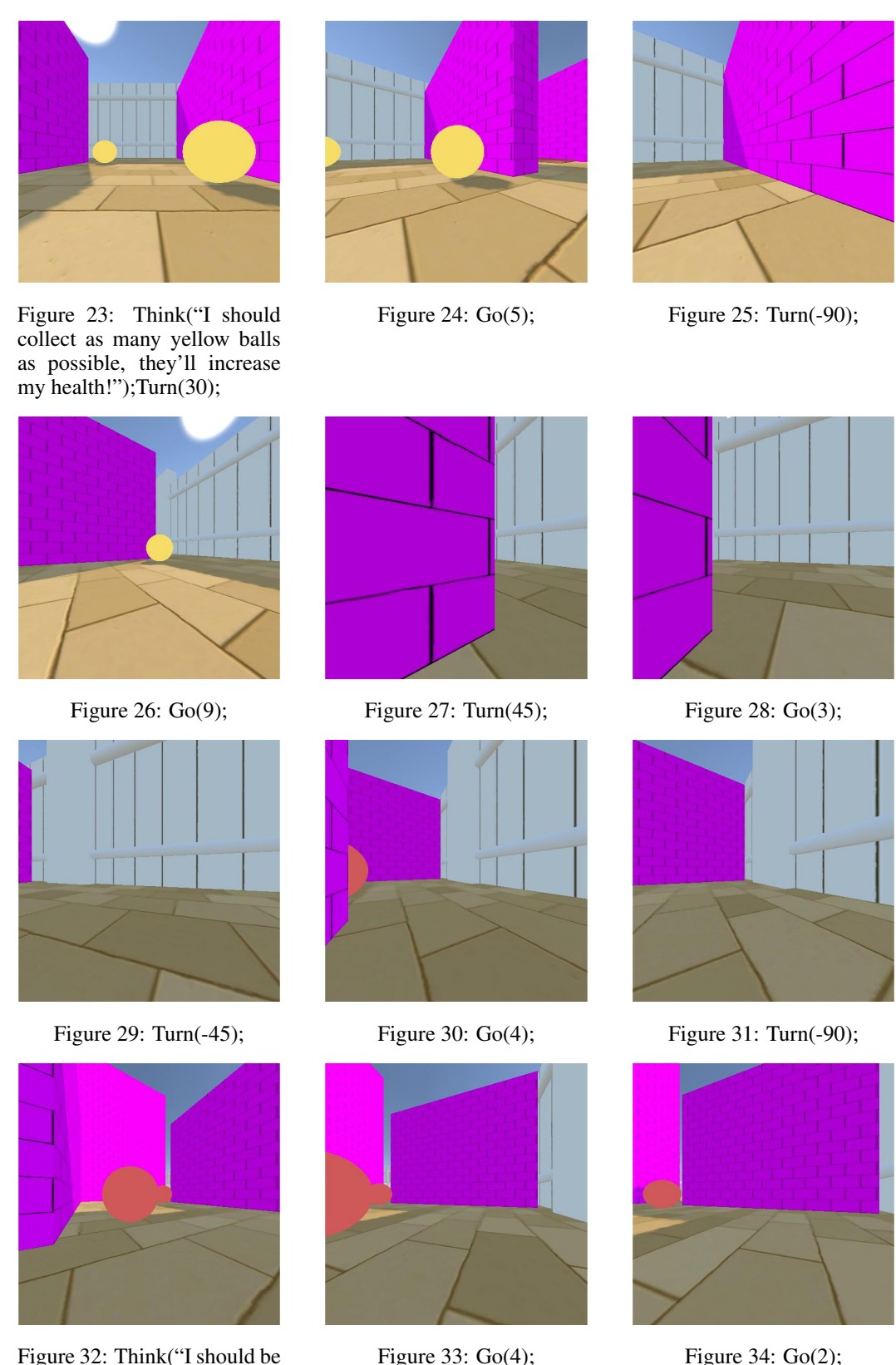

Figure 23: Think("I should collect as many yellow balls as possible, they'll increase my health!");Turn(30);

Figure 24: Go(5);

Figure 25: Turn(-90);

Figure 26: Go(9);

Figure 27: Turn(45);

Figure 28: Go(3);

Figure 29: Turn(-45);

Figure 30: Go(4);

Figure 31: Turn(-90);

Figure 32: Think("I should be very careful to avoid the red balls: if I touch one of those I'll lose the level!");Turn(45);

Figure 33: Go(4);

Figure 34: Go(2);

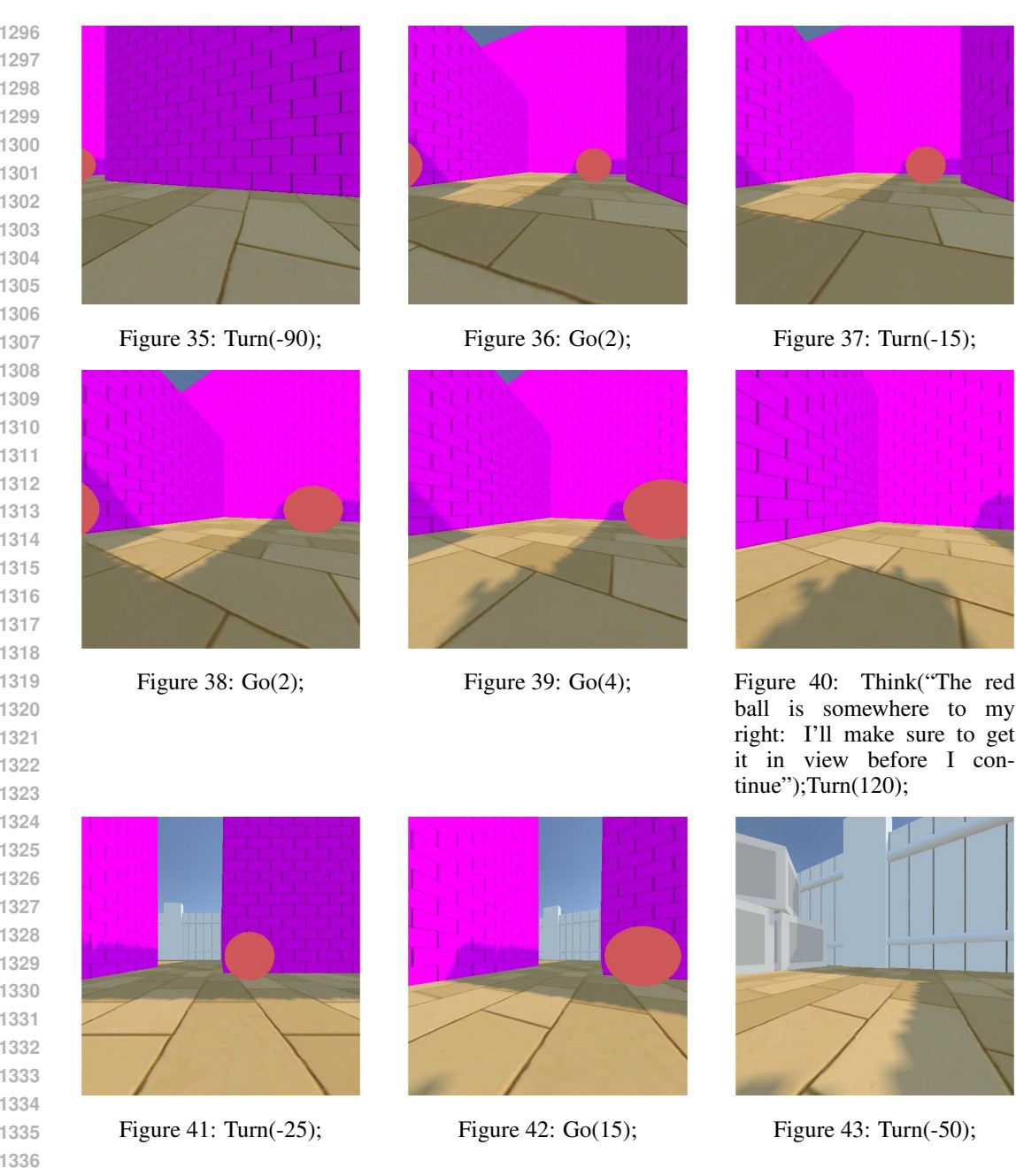

Figure 35: Turn(-90);

Figure 36: Go(2);

Figure 37: Turn(-15);

Figure 38: Go(2);

Figure 39: Go(4);

Figure 40: Think("The red ball is somewhere to my right: I'll make sure to get it in view before I continue");Turn(120);

Figure 41: Turn(-25);

Figure 42: Go(15);

Figure 43: Turn(-50);

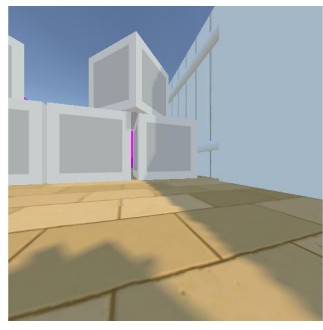

Figure 44: Think("There seem to be piled boxes in front of me: I'll push them out of the way by crashing into them with speed"); Go(15);

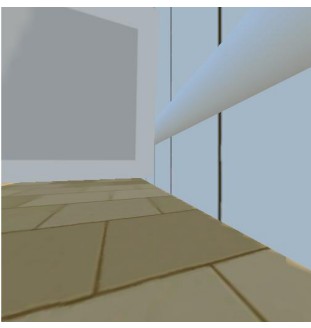

Figure 45: Think("I should confirm that I was successful in moving the boxes by turning around"); Turn(-120);

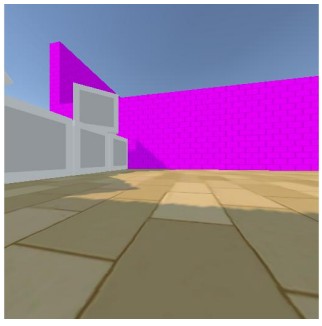

Figure 46: Think("I can see some piled boxes from the other side, so I have made it through. I'll turn to search this area for the reward"); Turn(90);

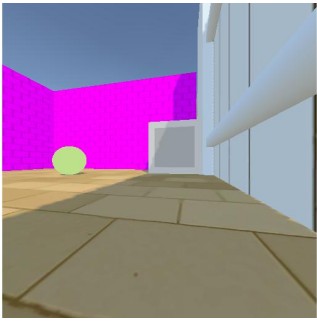

Figure 47: Think("The green ball is in view, I should turn about 30 degrees to my left to get it"); Turn(-30);

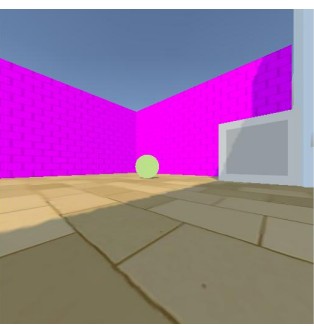

Figure 48: Think("The green ball is centered in my field of vision! I can advance forward to get it!"); Go(10);

