# OpenReview forum: "A little less conversation, a little more action, please: Investigating the physical common-sense of LLMs in a 3D embodied environment"
_ICLR.cc/2025/Conference — Submitted to ICLR 2025_

### Official Review · Reviewer_2hbs · 2024-10-28

**Soundness:** 2
**Presentation:** 3
**Contribution:** 3
**Rating:** 5
**Confidence:** 4

**Summary:**

This paper introduces LLM-AAI, a framework to evaluate Large Language Models (LLMs) on physical common-sense reasoning in a 3D embodied environment using the Animal-AI Testbed. This method allows for dynamic interaction, testing LLMs’ spatial and physical reasoning through tasks involving navigation, tool use, and object tracking. The study shows that state-of-the-art LLMs perform well on basic tasks but lack robust physical reasoning compared to human children and top reinforcement learning agents. While valuable for assessing LLMs’ spatial reasoning, the benchmark lacks components directly aimed at improving model training, functioning solely as a testing benchmark.

**Strengths:**

- Novelty: The LLM-AAI framework introduces a new approach by situating LLMs within 3D environments to evaluate physical common-sense reasoning, shifting the focus from static benchmarks to embodied assessments.
- Significance: This framework could set a new standard for LLM evaluation, addressing key questions about their ability to reason about and act in physical environments.
- Clarity: The paper is generally well-organized, with a logical flow from problem setup to experimental results and implications.

**Weaknesses:**

- The “Think” function, while useful, feels a bit too specific for genuine decision-making. Depending on the agent design, LLMs could have more diverse commands—like “use tools” or “save to memory”—to handle different situations. The decision to define the action protocol with just “Go,” “Turn,” and “Think” needs some justification. In other words, what is the design principle behind this protocol?
- The benchmark is also set up to test out-of-the-box LLMs, yet the comparison to children can feel unbalanced since LLM agents might benefit from a more complex setup than just these three actions.
- While this benchmark does a great job assessing spatial reasoning, it’s solely focused on testing. There’s not much clarity on how the framework could directly support model training or help develop improved physical reasoning abilities in LLMs.

**Questions:**

Overall I think the paper is proposing a benchmark for an important problem. However, I would like to hear the authors' opinions on the design principles of how LLMs interact with the environment.

---

> ### Author Response · Authors · 2024-11-26
>
> We would like to thank Reviewer 2hbs for their review. Please find below our responses.
>
> Our intention in this paper is to present a new methodology for evaluating Language Models, and the results of applying such a methodology to current state-of-the-art models. We thank the reviewer for noting that the approach is “valuable for assessing LLMs’ spatial reasoning”, and that it ”could set a new standard for LLM evaluation, addressing key questions about their ability to reason about and act in physical environments”, as well as their comments about the novelty and clarity of our work.
>
> > ‘The benchmark is also set up to test out-of-the-box LLMs, yet the comparison to children can feel unbalanced since LLM agents might benefit from a more complex setup than just these three actions.’
>
> We selected the actions and commands given to LLMs to be analogous to those given to children. Children use keys to move forward, backwards, left, or right, and they arguably think and plan their next steps, even if only implicitly. Likewise, including additional actions for the LLMs to use would have provided them with an advantage not available to the children. We have adopted this simple setup to allow for the fairest possible comparison between children and LLMs.
>
> > ‘There’s not much clarity on how the framework could directly support model training or help develop improved physical reasoning abilities in LLMs.’
>
> The goal of this paper was not to directly support model training but to create a new framework to evaluate LLM agents on commonsense spatial reasoning. However, one of the observations from the experiments was that LLMs struggled with navigation, and a further follow-up would be to fine-tune these models on simple navigation tasks. We have discussed this potential future direction in the “Limitations and Future Work” subsection. The reviewer may also be interested in our new section Appendix A, in which we discuss selected case studies from our experiments, which might provide greater insight into the underlying mechanisms governing LLM performance. These observations could also help to inspire new research directions for model development.

---

> ### Author Response · Authors · 2024-11-26
>
> >'Overall I think the paper is proposing a benchmark for an important problem. However, I would like to hear the authors' opinions on the design principles of how LLMs interact with the environment.'
>
> Three principles we would like to highlight that shaped our design of the environment interface are:
>
> - **The interface should be accessible to most researchers:** This precluded us from choosing options such as having the models choose an action for each frame in the environment, which would currently be prohibitively expensive for many researchers, and could present challenges to all but the models with the longest context windows.
>
> - **There should be a minimal chance of confounds from the interface:** As the reviewer highlights in their second point in the “Weaknesses” section, complex setups for agents may introduce confounds that make the comparison to children unfair. For this reason we chose to prioritise simplicity in the environment, avoiding the complex setups found in other papers, which may confound the comparisons we would like to make. We hope this helps to illustrate why we made the decision not to include actions such as  “use tools” or “save to memory”, as the Reviewer suggests including such commands would make it harder to understand which capabilities are possessed robustly by the model, and would make comparisons with other 'kinds of intelligence' such as children more difficult.
>
> - **The interface should be aligned with the current literature:** We adopted the ReAct framework to improve the LLM's decision-making process (Yao et al. 2022). The ReAct approach combines reasoning and acting by allowing the model to generate reasoning traces alongside actions (i.e. commands Go and Turn), which has shown improved performance on agentic tasks. By integrating ReAct, we encourage the LLM to first reason about the environment, identifying visible objects and their spatial relationships relative to the agent, before producing action scripts, such as Turn(15) and Go(10).

---

> > ### Comment · Reviewer_2hbs · 2024-11-27
> >
> > First of all, I agree with the principle that there should be minimal chances of confounding factors. However, the design of the three actions (“Go,” “Turn,” and “Think”) feels somewhat unnatural and unbalanced. Specifically, “Go” and “Turn” are highly specific to this particular environment, while “Think” is overly generic.
> >
> > If we take this approach, for example, in an environment where the agent needs to nail something, a “Use Tool” action could be added to better evaluate physical common sense. Similarly, a “Jump” action might be essential in an environment requiring the agent to traverse obstacles. These actions are also closely tied to the type of physical reasoning that this paper aims to assess.
> >
> > Additionally, the “Think” action appears overly generalized, with no clear justification as to why it should be “Think” but not others. For instance, an action like “Memorize”, "Sense", or some other commands could be just as relevant in reflecting the agent’s capacity for physical common sense.
> >
> > Overall, the design of these actions seems somewhat arbitrary. **Instead of being grounded in a clear and comprehensive set of principles for evaluating physical reasoning, the choices appear to be driven by the constraints of the AAI environment and the ReAct agent framework**. A more principled approach to designing the action set—one that systematically reflects the components required for physical reasoning—would significantly enhance the framework’s robustness and relevance.

---

> > > ### Author Response · Authors · 2024-11-30
> > >
> > > We thank the reviewer for engaging with our responses, and are pleased to hear that we share a common belief in the principle of trying to reduce the possibility of confounding factors. Overall, we would challenge the reviewer’s suggestion that ‘“Go” and “Turn” are highly specific to this particular environment’ - this is the space of movements available to a non-flying non-jumping creature in a 3D environment. The reviewer’s highlighted concern was that we were too driven by the choices of AAI and ReAct. Please see our responses to each below:
> > >
> > > **AAI environment**
> > >
> > > It is true that we’re constrained by the AAI environment, as any experiment is limited by the constraints of its laboratory. The constraints of the AAI environment have been designed with the principle of trying to reduce confounding factors in mind. We can discuss the possibilities of “jump” and “use tool” actions suggested by the reviewer as examples of this principle in action:
> > >
> > > - Our shared principle of reducing potential confounding factors would indicate that it is safer to minimise the potential actions the agent can take, subject to the environment being sufficiently expressive. 'Go' and 'Turn' already enable a vast space of interactions with the environment, especially when considering interactions with objects. The rich space of possible interactions has allowed the creation of large cognitive test sets designed in AAI (see e.g. Voudouris et al., 2024). This, combined with the fact children and Competition (RL) agents comfortably outperform LLMs, suggests that AAI is sufficiently expressive. Therefore, with our principle in mind we would suggest that actions such as "jump" should be excluded until there is a strong case for their inclusion (such as a concrete example of a cognitive capability that cannot be tested without it).
> > >
> > >     To make our application of this principle more specific, there are reasons why we might expect "jump" to be the source of confounds. For example, adding an extra "jump" action adds additional complexity in planning navigation, which would be an additional hurdle for LLMs to overcome. Or the "jump" action may be the source of an unintended "shortcut", in which instead of solving a task with the facet of physical reasoning we’re trying to test, an agent might find a way of using the jump action to bypass the challenge set for it.
> > >
> > > - A “use tool” action would remove many of the interesting cognitive challenges of tool use (in the paper we refer to these as functions that “outsource cognitively interesting tasks”). A ramp can be seen as a very simple tool to “go up”; in our revised paper we added a discussion of how all the LLMs failed to use this tool to achieve their goal (see appendix A.1). If the LLMs had had access to a “use ramp” action, we would have lost this insight into their shortcomings. In the reviewer’s example of nailing something, without “use hammer” an agent would need to recognise that a hammer is a hammer, that it must be swung to be used, that it can only be used from one end, and so on. These are exactly the capabilities we’re trying to test by excluding more high level functions such as “use tool”.
> > >
> > > **ReAct framework**
> > >
> > > The reviewer mentions “Memorize”, or "Sense" as possible actions to be included, but we would consider these to be features of the agent and not features of the environment. As with the “use tool” example above, we don’t want to provide any actions that may take on the cognitive challenge for the agent (note that the children assessed on these tasks had to bring their own memorisation and sensing capabilities).
> > >
> > > On the other hand, the “Think” action allows LLMs to freely express any type of cognitive content necessary, including (but not limited to) noting observations, recalling prior information, and devising action plans. Also, it reflects a very common usage pattern in LLMs, and so its inclusion follows our principle that “The interface should be aligned with the current literature”. This usage pattern is encapsulated in the ReAct framework for agents, but also more broadly in the trend of asking LLMs to “explain their reasoning” to improve performance. If and when LLMs progress to the level where this pattern is no longer required (OpenAI’s o1 appears to be moving in this direction, by internalising it in the model) a future LLM-AAI could potentially do away with the “Think” action, further simplifying the framework.
> > >
> > > We enjoyed receiving the reviewer’s comments, which reflect many of the discussions that we have had during the design of LLM-AAI and related work. We would be excited to hear if they are satisfied by our comments, or to continue the discussion.

---

### Official Review · Reviewer_772C · 2024-10-29

**Soundness:** 1
**Presentation:** 1
**Contribution:** 1
**Rating:** 3
**Confidence:** 2

**Summary:**

This study presents a framework for assessing physical common-sense reasoning in large language models (LLMs) using the Animal-AI (AAI) environment. By replicating animal lab experiments in a virtual 3D lab, the framework compares LLMs with Deep Reinforcement Learning agents, humans, and animals on tasks like distance estimation, obstacle navigation, object tracking, and tool use.

**Strengths:**

It builds a platform and benchmark for physical common-sense reasoning evaluations. In this platform, rather than reasoning, the agent can make actions to show their understanding of the environment.

**Weaknesses:**

1. Throughout the paper, the authors refer to using LLMs, though VLMs would be more accurate, given that the observations are primarily image-based.

2. The environment appears overly simplistic, lacking reflections of real-world physics and focusing mainly on semantic aspects.

3. The central contribution of this work remains unclear, as much of the effort seems concentrated on API and prompt design.

**Questions:**

The major contributions of this paper are unclear to me, leading to the following questions:

1. What real-world physical phenomena are represented in this environment?
2. How does the work demonstrate that the VLMs can reason about these physical phenomena?
3. What are the primary challenges in building such a benchmark, and how does this paper address them?
4. Are there any comparable previous works? What specific advantages does this work offer over them?

---

> ### Author Response · Authors · 2024-11-26
>
> We would like to thank Reviewer 772C for their review. Please find below our responses.
>
> The reviewer’s summary of our paper suggests we compare the results of LLMs with the performance of animals. While the Animal-AI environment could be used to make such comparisons (either by conducting the same experiment in Animal-AI and in the physical laboratory, or by using VR) we do not make such comparisons in the paper.
>
> We thank the reviewer for their suggestion of an improvement to the terminology of our paper. There is some variation in terminology in the literature. We have now stated in the introduction that the subject of the paper is specifically multi-modal LLMs. A reason we do not want to limit ourselves to the term ‘VLMs’ is that the LLM-AAI framework could be extended to generate purely textual descriptions of observations of the environment, meaning that text-only LLMs would be able to interact with it. We performed some initial experimentation along these lines, but due to the limited performance of these models, we do not present results from them in the paper. Nevertheless, this is an option available for future research if and when text-only LLM capabilities improve in this domain.
>
> We would challenge the reviewer’s suggestion that the environment lacks reflections of real-world physics. The Animal-AI environment is built using the Unity physics engine. While the shapes and textures of objects are minimalist in cosmetic details, they behave like objects in the real world in terms of their physics. All objects in Animal-AI have masses, volumes, and friction coefficients, are subject to the pull of earth-like gravity, and display realistic inertia and momentum. We have included a discussion of these points in Section 3.
>
> We were unsure what the reviewer meant by “focusing mainly on semantic” aspects. Would the reviewer be able to clarify what they meant here, and how we might broaden the focus of LLM-AAI to address their concerns?
>
> We view our central contribution of the paper to be the presentation of a framework for studying physical common-sense reasoning in LLMs, using well-described and validated experiments from comparative cognition. Furthermore, we present evidence that state-of-the-art multimodal LLMs lack physical common-sense reasoning, although they are able to interact effectively with an environment on simple tasks. As reviewer rqxH notes, this “methodology’s simplicity and focus make it easy to interpret and isolate the abilities being measured”. This is echoed by reviewer 2hbs, who notes that LLM-AAI is “valuable for assessing LLMs’ spatial reasoning”.

---

> ### Author Response · Authors · 2024-11-26
>
> >'What real-world physical phenomena are represented in this environment?'
>
> As the environment is built using the Unity Physics engine, it provides fast and realistic physical dynamics. In particular, all objects in Animal-AI have masses, volumes, and friction coefficients, are subject to the pull of earth-like gravity, and display realistic inertia and momentum. Note that the environment only represents physical dynamics to the extent that it allows researchers to test physical common sense reasoning. Animal-AI has already been used extensively to study physical common-sense reasoning in reinforcement learning agents and humans, and there is a growing body of experiments using the environment that replicate experimental designs used in real-world laboratories with animals and children.
>
> >'How does the work demonstrate that the VLMs can reason about these physical phenomena?'
>
> Our work does not demonstrate that the VLMs can reason about these physical phenomena, and provides evidence through their failure that they lack the ability to perform such reasoning. This highlights a significant shortcoming in current systems.
>
> Were a model to achieve good performance on these tasks, it would provide stronger evidence of their ability to perform such reasoning than current alternative evaluations because (1) The tasks in AAI draw on the history of experimentation in comparative cognition to ensure they are robust and focused on different facets of reasoning, and (2) The AAI environment requires LLMs to respond dynamically to the demands of a simulated 3D environment.
>
> >'What are the primary challenges in building such a benchmark, and how does this paper address them?'
>
> Generally, we would summarise the primary challenges of building benchmarks as (1) ensuring they precisely and comprehensively measure the capability(s) of interest, and (2) ensuring the results are informative and actionable, identifying when and where models are successful and what their key failure modes are. LLM-AAI addresses the first point by providing a robust and well-validated framework for studying physical common-sense reasoning in multi-modal LLMs. It is an improvement on the current state-of-the-art which is to use Q&A-style benchmarks, because LLM-AAI uses a rich and veridical 3D environment and is less easily gamed or contaminated (models cannot be trained on all the state-action pairs present in AAI, whereas they can be for static text-and-image benchmarks). LLM-AAI addresses the second point by covering a variety of physical common-sense reasoning tasks using validated experiments from cognitive science. This allows us to precisely identify the kinds of tasks that contemporary LLMs can solve and those in which they fail, thus informing future work on developing better training curricula and model architectures. Moreover, AAI facilitates the iterative development of new experimental designs, leading to deeper insights about system capabilities.
>
> The Psychology literature provides a rich source of inspiration for the kinds of capabilities we should be evaluating in machine learning systems and how we should do so, but it is currently a challenge for machine learning researchers to benefit from this as it is a distinct body of work which few of them will be familiar with. LLM-AAI addresses this challenge by providing a common space in which researchers from Psychology can collaborate with researchers developing Language Models for the benefit of both fields.
>
> >'Are there any comparable previous works? What specific advantages does this work offer over them?'
>
> We summarise the state-of-the-art in physical common-sense evaluation in LLMs in Section 2. Most closely related to LLM-AAI is work employing VirtualHome to study ‘LLM Agents’ (e.g., Huang et al., 2022; Xiang et al., 2024; Li et al., 2024). However, the focus of these environments is to evaluate whether LLMs can follow instructions when interacting with an environment. They do not seek to directly measure the physical common-sense reasoning of these systems. This is the niche that LLM-AAI fills.

---

> ### Author Response · Authors · 2024-11-27
>
> We thank the reviewer for raising their rating from a 1 to a 3. We would be pleased to discuss any remaining concerns about our paper: Please could the reviewer describe which of their concerns are still unaddressed?

---

> > ### Comment · Reviewer_772C · 2024-11-28
> >
> > My major concern is still for the contribution part. I don't think the methodological contribution could serve alone as the main contribution of a paper. However, I am not sure if this is also the standard of the community. That's why my confidence is 2 in this assessment.

---

> > > ### Author Response · Authors · 2024-11-29
> > >
> > > We thank the reviewer for explaining where their doubts lie. The [ICLR call for papers list](https://iclr.cc/Conferences/2025/CallForPapers) of relevant topics includes 'infrastructure, software libraries, hardware etc'. What we've done is created an infrastructure (a framework) that allows us to test multimodal LLMs on AAI. We're also tapping into the 'datasets and benchmarks' topic as we're making AAI (a benchmark in the general sense) available for testing and benchmarking multimodal LLMs.
> > >
> > > Based on these features of the call for papers, we would ask the reviewer to reconsider their suggestion that a methodological contribution cannot be the main contribution of a paper. We hope our clarification satisfies the reviewer’s concern, but are happy to discuss further.

---

### Official Review · Reviewer_rqxH · 2024-11-03

**Soundness:** 2
**Presentation:** 2
**Contribution:** 2
**Rating:** 5
**Confidence:** 3

**Summary:**

This paper presents a framework, LLM-AAI, to assess the physical common-sense reasoning abilities of large language models (LLMs) by situating them within a 3D virtual environment known as Animal-AI. The authors aim to evaluate how well LLMs perform tasks such as navigation, object permanence, and tool use by using a suite of tests inspired by cognitive science. The framework enables direct comparisons between LLMs, reinforcement learning (RL) agents, and human children. The results show that LLMs struggle with complex physical reasoning tasks, often underperforming compared to both children and specialized RL agents.

**Strengths:**

The paper uses a minimalist yet structured environment (Animal-AI) that allows researchers to test LLMs’ physical common-sense reasoning in a controlled setting. This setup avoids the limitations of static benchmarks by providing a dynamic, interactive arena where LLMs can showcase their reasoning abilities in real-time. The use of Animal-AI as a testing ground allows the study of fundamental aspects of physical common sense, enabling a valuable comparison between the performance of LLMs and other entities, including human children. The methodology’s simplicity and focus make it easy to interpret and isolate the abilities being measured.

**Weaknesses:**

A significant limitation of this paper lies in the simplicity, or “toy” nature, of the Animal-AI environment. Although useful for initial assessments, the constrained setup does not approximate the complexities of real-world interactions. There is an inherent risk that high performance in this environment could falsely suggest deeper physical understanding, a trend that has been observed in prior research on similar “toy” environments. More robust environments, such as Minecraft or Omniverse, may better reveal practical competencies in physical reasoning due to their greater fidelity to real-world dynamics.

Furthermore, the paper lacks novel insights; it primarily documents performance differences without offering deeper interpretations or mechanisms underlying these differences. As such, it provides limited understanding beyond comparative scores, which raises questions about the broader applicability and practical utility of the results.

**Questions:**

What can the community learn from this work, apart from performance difference?

---

> ### Author Response · Authors · 2024-11-26
>
> We would like to thank Reviewer rqxH for their review. Please find below our responses.
>
> Firstly, we were pleased to see the reviewer’s acknowledgement of our approach’s benefits over the limitations of traditional, static benchmarks; we believe that our contributions will help to progress the science of evaluation of LLMs beyond its current static benchmark-centric state.
>
> > ‘A significant limitation of this paper lies in the simplicity, or “toy” nature, of the Animal-AI environment. More robust environments, such as Minecraft or Omniverse, may better reveal practical competencies in physical reasoning due to their greater fidelity to real-world dynamics.’
>
> We would push back against the claims that the Animal-AI environment is a toy environment that is less robust and has less fidelity than for example Omniverse. The Animal-AI environment is built using the Unity physics engine. While the shapes and textures of objects are minimalist in cosmetic details, they behave like objects in the real world in terms of their physics. All objects in Animal-AI have masses, volumes, and friction coefficients, are subject to the pull of earth-like gravity, and display realistic inertia and momentum. The engine is powerful and expressive, like that used in Omniverse, and is more physically realistic than the physics of Minecraft. We have added a discussion of these points in Section 3.
>
> We agree with the reviewer that results in overly simplistic environments can sometimes be misleading. Our approach is centred around a ‘virtual laboratory’ — the Animal-AI environment — where we strive to find the balance between having too simple an environment (which could present too easy a challenge and lead to false confidence in a model’s capabilities), and having too complex an environment (which could introduce additional demands that are irrelevant to the capability being investigated). Indeed, following best practices in experimental design, we seek to maximise the complexity of the environment in ways that allow us to precisely test the capability of interest (physical common sense) and minimise its complexity in ways that would obfuscate that capability (photorealistic detailing of objects, a large action space, etc.). We believe that our results indicate that Animal-AI strikes the right balance for the current wave of systems; nontrivial challenges in the environment are very difficult and unlikely to be solvable by LLMs in the near future, and yet, in the spirit of laboratory research, our approach isolates the capabilities of interest (e.g., object permanence, causal reasoning, numerosity), without adding unnecessary complexity. While Omniverse and Minecraft could offer an alternative for building these experiments, we disagree that they offer “greater fidelity to real-world dynamics”. Moreover, their complexity and, in the case of Minecraft, highly stylised design, would introduce further confounds into experimental design. Finally, the Animal-AI environment is specifically designed for conducting behavioural experiments drawn from cognitive science — Omniverse and Minecraft are not.
>
> We would argue that the performance differences are themselves part of our novel insights; we provide the first results for physical common sense in LLMs that can take actions in an environment, alongside comparisons with humans and reinforcement learning agents.
>
> In addition to these results, we believe that our key novel contributions are the benchmark and the LLM-AAI framework. We hope that our benchmark and baseline results will be used to evaluate future generations of LLMs, fuelling the development of LLMs with greater physical common sense; a capability which many embodied agents require. Moreover, we hope that our framework, which will be open-source, will serve the research community as both a theoretical template and a usable tool for testing the embodied cognitive capabilities of future LLMs and other multimodal systems.
>
> We agree with the reviewer that further interpretations of the LLMs’ performance and behaviour would improve our paper. We have taken these comments on board and have added a new section (Appendix A) where we highlight several key insights from the LLMs’ verbal and action responses that may shed some light on potential underlying mechanisms, and could provide a springboard for future work.

---

> ### Author Response · Authors · 2024-11-26
>
> > 'What can the community learn from this work, apart from performance difference?'
>
> In addition to the performance difference between child and LLM data, the community can also take two methodological lessons from this paper. Firstly, we have demonstrated how, instead of using static benchmarking to learn broad lessons about the task performance of LLMs, we can draw upon the rich history of experimentation in comparative cognition (the study of behaviour in non-human animals) to collect fine-grained data about LLM capabilities, and to make direct comparisons with other ‘kinds of intelligence’, such as humans and reinforcement learning agents. Secondly, we have demonstrated how simulated environments can be used as an alternative to traditional benchmarks in LLM evaluation.
>
> Both of these lessons have been articulated to some extent in other works, but we believe that our work is original in combining them. We have suggested that the Animal-AI environment is well suited to putting these lessons into practice.

---

> > ### Comment · Reviewer_rqxH · 2024-12-02
> >
> > I'd like to thank the authors for answering my questions.
> >
> > The environment may not be as toy as I expect. However, my major concerns on insights and contributions are not addressed. And I keep my initial rating.

---

### Author Response · Authors · 2024-11-26

We thank the reviewers for their feedback and suggestions. Through reading the reviews, it became clear that we were insufficiently clear in some areas, particularly in conveying that our main contribution is methodological—the creation of the LLM-AAI framework for testing embodied cognition in LLMs. Additionally, we recognise the need to better explain the Animal-AI environment, which was carefully designed to balance complexity and simplicity. This balance allows for robust evaluation while avoiding unnecessary noise and confounds. Below, we summarise the key contributions and responses.

Our paper introduces the LLM-AAI framework, a novel methodology for evaluating the physical common-sense reasoning of multimodal LLMs. Unlike traditional static benchmarks, LLM-AAI incorporates validated experiments from comparative cognition within the Animal-AI environment, a 3D simulation platform specifically designed for precise testing. Powered by the Unity physics engine, the environment provides realistic physical dynamics (e.g., gravity, inertia) while avoiding excessive visual complexity or the confounding factors seen in environments like Minecraft or Omniverse. This careful design ensures that evaluations focus specifically on reasoning capabilities such as object permanence and causal reasoning, without being distracted by irrelevant complexities. We have clarified these points in Section 3 of our paper.

Our results highlight significant deficiencies in the physical common-sense reasoning abilities of state-of-the-art multimodal LLMs. By providing a reusable and open-source framework, we aim to enable future research to address these shortcomings and improve LLMs for embodied tasks. Additionally, our work bridges interdisciplinary gaps by offering a methodology inspired by psychology and comparative cognition, fostering collaboration across fields, and extending LLM evaluation beyond static benchmarks.

Finally, we have included a new set of qualitative results highlighting the reasoning patterns driving LLMs when planning and acting in the environment (Appendix A). These results complement our findings from Experiments 1 and 2 and point toward fruitful directions for future research.

---

### Meta-Review · Area_Chair_xHwT · 2024-12-11

**Metareview:**

This paper introduces LLM-AAI, a framework for evaluating the physical common-sense reasoning abilities of Large Language Models (LLMs) within the Animal-AI 3D virtual environment. The framework examines tasks such as navigation, object permanence, and tool use, drawing inspiration from cognitive science research. LLM-AAI enables direct comparisons between LLMs, reinforcement learning agents, and human children through tasks designed to be dynamic and realistic. Results show that while LLMs perform reasonably well on simpler tasks, their performance is inconsistent, particularly on tasks requiring dynamic reasoning, and they significantly underperform on complex physical reasoning compared to both human children and RL agents.

Reviewers appreciated the dynamic, interactive arena where LLMs can showcase their reasoning abilities in real-time. They also appreciated the overall presentation and clarity. However, they raised concerns about the simplicity of the Animal-AI environment, describing it as a toy setup that lacks the complexity needed to approximate real-world interactions. They noted that the paper focuses on documenting performance differences but does not delve into deeper insights or mechanisms explaining these differences. Furthermore, the benchmark serves solely as a testing framework, offering no direct pathways for improving LLMs’ physical reasoning abilities.

In the post-rebuttal discussion phase, reviewers converged on the view that the paper provides limited technical contributions and insights to the community. After careful examination and discussions with reviewers, the AC finds the arguments from reviewers `R#rqxH` and `R#2hbs` to be the most significant in evaluating this submission. While the authors provided a response, it did not sufficiently address the concerns raised, and reviewers remained unconvinced to support acceptance at ICLR.

**Additional Comments On Reviewer Discussion:**

- The AC finds the arguments from reviewers R#rqxH and R#2hbs (both scoring 5) to be the most significant in evaluating this submission. Both reviewers engaged in the post-rebuttal discussion phase.

- Reviewer R#772C (score changed from 1 to 3) concurred with the concerns about limited novelty but also mentioned their reduced confidence, which the AC appropriately weighted.

- All reviewers acknowledged and responded to the rebuttal, but the concerns remained insufficiently addressed to support acceptance at ICLR.

---

### Decision · Program_Chairs · 2025-01-22

Reject